# Symptomatology of Long COVID Associated with Inherited and Acquired Thrombophilic Conditions: A Systematic Review

**DOI:** 10.3390/v17101315

**Published:** 2025-09-28

**Authors:** Amelia Mae Heath, Dan Li

**Affiliations:** 1British International School of Chicago, South Loop, 161 W 9th St, Chicago, IL 60605, USA; 2Massachusetts General Hospital, 55 Fruit Street, Boston, MA 02114, USA

**Keywords:** long-COVID, post-acute sequelae of SARS-CoV-2, thrombophilia, microthrombi, COVID-19, coagulation, SARS-CoV-2, thrombosis

## Abstract

Thrombophilic conditions, conditions where blood has a tendency to form thrombi due to abnormal coagulatory processes, can affect the trajectory of diseases such as Post-Acute Sequelae of SARS-CoV-2 Infection, better known as Long COVID (LC), by worsening symptoms and complicating outlooks. As a comorbidity in pro-coagulatory diseases such as COVID-19 and LC, patients with thrombophilic conditions may experience worse symptoms than their peers, due to this elevated level of hypercoagulation. A 15-week literature review through the public PubMed database was conducted to investigate the severity, mechanisms, and symptom profiles of thrombophilic patients with LC. Papers were only included if samples included participants with pre-existing tendencies for hypercoagulable states, and confirmation of SARS-CoV-2 infection via a Polymerase Chain Reaction test. Each paper included in this review was analyzed by topic and assessed for eligibility against the Joanna Briggs Institute’s Critical Appraisal tool. Each paper was also assessed for biases. Results from the 6 papers included in this review showed that LC could be predicted following COVID-19 illness by a hypercoagulable blood profile, indicating that LC may be linked to chronic hypercoagulation and inflammation post-infection. Additionally, symptoms linked to microthrombi formation, such as hair loss, arrhythmia, and dizziness, were exhibited more frequently in patients with thrombophilia and/or thrombophilic conditions, indicating that those with thrombophilic conditions may exhibit unique LC symptom profiles compared to healthy controls. This paper’s research is preliminary and thus is limited in the strength of its findings; However, further research into LC and its interactions with co-morbidities like thrombophilic conditions would aid in the development of better treatment plans for patients, such as the usage of anticoagulants or screening for hypercoagulable blood profiles post-COVID-19 to assess patient risk.

## 1. Introduction

A hypercoagulable state, also known as thrombophilia, is a pathological state in which one’s blood is prone to blood clotting. As a result, thrombophilia is associated with an increased risk of the formation of thrombi within blood vessels, often undetected due to general manifestations without clinical symptoms, which can lead to severe complications, such as strokes or venous thromboembolisms, based on the location of the clot [1]. This state can be temporary, or can be a long-term condition, either through inheritance (also known as primary thrombophilia/hypercoagulable states) or through its acquisition via a variety of different disorders and conditions (also referred to as secondary thrombophilia/hypercoagulable states) [2]. The origin of thrombophilic conditions in patients is generally assessed via various screening tests for primary thrombophilic conditions such as Antithrombin III deficiency or Factor V Leiden mutations, along with a collection of familial history and personal risk factors [1]. In a sick individual, those in hypercoagulable states have a higher risk of developing blood clots, particularly in infections that elicit a strong inflammatory response [1], which is generally treated with the prescription of prophylactic anticoagulants. This is exemplified by the disease COVID-19, caused by the virus SARS-CoV-2, where microthrombi have an increased tendency to form compared to other diseases. These microthrombi can obstruct lumens and impair oxygen delivery to nearby tissues. Thus, patients with thrombophilic conditions are at a higher risk of hospitalization and developing severe COVID-19 infections. Furthermore, research has shown that blood clotting risk is elevated for up to a year past the initial contraction of the disease [3].

Long COVID (LC), also known as Post-COVID Conditions or post-acute sequelae of SARS-CoV-2, is an infection-associated chronic condition from the SARS-CoV-2 virus, lasting a minimum duration of 3 months. Symptoms can persist, recur, or newly appear 4 or more weeks following the initial infection, and can range from mild (such as “brain fog” or extreme tiredness) to moderate (such as irregular heartbeats or issues with digestion) [4,5,6]. The nature of LC physiological mechanisms is unclear; thus, current treatment for LC focuses primarily on the alleviation or minimization of symptoms instead of their removal. Though little is known about risk factors for the development of LC, it has been shown that those who experienced severe cases of COVID-19 have a higher likelihood of developing LC.

Since those in chronic hypercoagulable states are at a greater risk of developing severe COVID-19 cases, they are consequently more likely to develop LC. These patients are particularly vulnerable to severe complications, not only from their thrombophilic conditions but also from underlying conditions that have led to their hypercoagulable states [7]. The vulnerability of this population to develop the chronic condition of LC, combined with their pre-existing conditions, may lead to the development of more severe or alternate symptoms from this condition compared to the normal population. Due to the elevated risk of this subset developing LC, the effects of the condition should be studied. Currently, there is minimal research about the effects of chronic hypercoagulable states on the symptomatology of LC, despite the known links between COVID-19, abnormal blood clotting, and chronic conditions. Furthermore, there have been no previous review articles investigating the interactions between LC and thrombophilic conditions, with published articles discussing the topic being sporadic, often with a focus on individual cases or individual mechanisms: this further emphasizes the importance of this review to synthesize previous research to better comprehend the interaction of multiple mechanisms within LC pathology and thrombophilic conditions.

This systematic review aims to investigate the main question of whether LC patients in chronic hypercoagulable states are more likely to develop symptoms related to microthrombi. PubMed was used to search for existing research on conditions involving hypercoagulative states and LC symptoms, and after screening and analysis of each article, the collective findings were recorded. This review highlights the possible applications in the diagnosis of LC in patients with known hypercoagulative states, and may serve as guidance for the prevention of severe symptoms or the development of therapeutic suggestions in the medical field. Thus, this review aims to investigate the research question, “are LC patients in chronic hypercoagulable states more likely to develop symptoms related to microthrombi?”

## 2. Materials and Methods

This overarching question will be split into more specified subtopics. Some of the subtopics this systematic review hopes to explore under this main question include: To investigate if LC patients in chronic hypercoagulable states are more likely to develop symptoms related to microthrombi, data was collected from PubMed publications that explored research in: the prevalence of thrombophilia in LC patients; the symptomatology of chronic hypercoagulable LC patients; the potential mechanistic pathways of LC, related to thrombophilia; and the effect of primary (inherited) thrombophilia versus secondary (acquired) thrombophilia on LC symptomatology and development. We also decided to collect data from each of these papers on study design and identification, population characteristics, confounding variables and biases, and key takeaways as part of our results synthesis. We utilized the PRISMA checklist to ensure the transparency and credibility of this paper; however, this review is not registered in a database, nor is its protocol published.

### 2.1. Search Strategy

We conducted a systematic review of papers found via PubMed’s advanced search tool from 12 July 2025 to 22 July 2025. By designing Boolean operators, we created a search that only shows papers that included both terms relating to thrombophilia AND LC. The keywords are as follows:

Thrombophilia: “Hereditary Thrombophilia,” “Inherited Thrombophilia,” “Congenital Thrombophilia,” “Factor V Leiden,” “Leiden mutation,” “Factor V Mutation,” “Prothrombin G20210A,” “Prothrombin gene mutation,” “Protein C deficiency,” “Protein S deficiency,” “Antithrombin deficiency,” “AT III deficiency,” “Activated Protein C Resistance,” “APC resistance,” “Homocysteinemia,” “Hyperhomocysteinemia,” and “MTHFR mutation.”

Long COVID: “long COVID,” “Long haul COVID,” “COVID long hauler,” “Post-acute COVID-19 syndrome,” “post COVID syndrome,” “post COVID syndrome,” “Chronic COVID Syndrome,” “PASC,” “Post-acute sequelae of COVID-19,” “Persistent COVID symptoms,” and “Sequelae of COVID-19.”

Additionally, we filtered for papers produced only in English and texts that contained free full versions of the paper.

### 2.2. Screening Protocol

We conducted two rounds of screening—first, by screening the title and abstract, and secondly, by screening the full text of the manuscript retrieved by our search strategy. Figure 1 provides a visualized Preferred Reporting Items for Systematic Reviews and Meta-Analyses (PRISMA) diagram of our process.

### 2.3. Inclusion Criteria

Population: We did not exclude any papers based on population.

Concept: We excluded studies that did not include samples with pre-existing tendencies for hypercoagulable states. Papers with LC cohorts must have confirmed prior COVID-19 via a positive Polymerase Chain Reaction (PCR) test.

Context: We did not exclude any papers based on geographical context.

### 2.4. Data Extraction

For each paper, we extracted data about each subtopic and entered it into a spreadsheet. For the first paper, “Hypercoagulable Rotational Thromboelastometry During Hospital Stay Is Associated with Post-Discharge DLco Impairment in Patients with COVID-19-Related Pneumonia,” we completed this scanning manually, and then utilized the language-learning model (LLM) ChatGPT version 5.0 to confirm our extraction. By comparing the manual and generated responses to the subtopics, we were able to confirm the reliability of ChatGPT in extracting data. We used the same prompt each time to ensure replicability, which is shown below:

“Disregard any previous message or information collected. *

Following this message, I will attach a research paper. Extract information from solely this paper on each of the following topics, answering each of the questions listed in each topic:1.Study design and identification—origin of studies, authors, study designs, inclusion and exclusion criteria, timeframe of experiment, objective, and if ethical approval was gained (and by whom).2.Population characteristics—including age, sex, ethnicity, and comorbidities of the studied population.3.Prevalence of thrombophilic conditions in Long COVID patients—is there a larger proportion of long COVID patients with pre-existing thrombophilic conditions? Do those with said conditions have a greater risk of developing Long COVID?4.Symptom profiles of patients with thrombophilic conditions experiencing Long COVID—are certain symptoms more prevalent or severe in patients with thrombophilic conditions? Do these potentially relate to the formation of microthrombi?5.Potential mechanistic pathways of Long COVID and thrombophilia—could potential reasons for symptom profiles be explained from data, i.e., via Magnetic Resonance Imaging (MRI) results, causal pathways, evidence of microclots, etc.6.The effect of primary (inherited) thrombophilia versus secondary (acquired) thrombophilia on Long COVID symptoms and outlook—are certain symptoms exhibited mostly in a specific form of thrombophilia? Do medical treatments and outlooks differ between the groups?7.Confounding and biases of research—are there limitations within the research design, methodology, or sample? Were these properly controlled? How have confounders been adjusted for?8.Key takeaways—Do findings suggest a relationship between symptom profiles of thrombophilic patients and the development of microthrombi? Is there an impact on treatment/outcome in Long COVID due to thrombophilia?

Only extract information from this paper, and do not add information from other sources. Add information that was not included in the questions if it appears essential to understanding or is vital to extracting results accurately.”

* used only when completing multiple extractions from consecutive papers, to ensure the model was not using any data from past papers.

Results obtained from the LLM were manually confirmed by searching through the paper—if the LLM was found to be inaccurate, results were recorded based on statistical evidence in the paper into the structured data form to minimize bias. Temporary chat was used to prevent the model from updating to include information from previously inputted information. Table 1 shows the paragraph lengths of each of the LLM’s responses, and whether the extracted data was primarily quantitative or qualitative.

### 2.5. Quality Assessment

To assess the quality of the studies included within our review, we utilized the Joanna Briggs Institute’s Critical Appraisal tools as a quality assessment. Table 2 demonstrates our results.

Papers that scored 80% or above on the JBI Critical Analysis scale were considered strong sources, and studies between 60–79% were considered middle quality, whereas studies that scored under 60% were considered low quality. From this quality assessment, we can confirm that all of the studies included within this review were of an acceptable quality to include within our review.

## 3. Results

### 3.1. Selection of Included Papers

Our PubMed search elicited thirteen total papers, none of which were duplicates. Six papers were excluded based on our inclusion criteria (detailed in Section 2.3), resulting in seven texts. We then sought retrieval of the full articles, one of which was inaccessible due to subscription restrictions, and was thus excluded from our review. Each of these texts then underwent final revision based on our inclusion criteria for inclusion within our systematic review. As all of these papers satisfied our inclusion criteria, they were included within our final review, resulting in six total studied papers.

### 3.2. Study Design and Identification

This topic is meant to synthesize the design of the papers reviewed, and focuses on various factors such as origin, inclusion & exclusion criteria, experimental timeframe, ethical approval, and objectives of the papers reviewed to best understand and address commonalities and shortcomings within study design. The papers reviewed in this article consisted of three narrative review articles [9,10,11], one prospective observational study [12], one longitudinal prospective observational study [13], and one cross-sectional and cohort study [14]. None of the papers had overlapping authors, and all studies obtained ethical approval. Of the three experiments, one was conducted in Athens, Greece [12], and two were conducted in Brazil: one in São Paulo [13] and the other in Belém [14]. Of the reviews, one was conducted in Madrid, Spain [9], one was conducted in Krakow, Poland [10], and one was conducted in the USA [11]. Across all the studies, data were collected between July 2020 and March 2022, with the median length of experimentation being 11 months, though this varied widely. All studies required participants within the COVID-19 cohorts to have tested positive on PCR for SARS-CoV-2 infection. Two of the studies [12,14] only studied patients with mild COVID-19 (which was defined as not requiring hospitalization), but one study [13] also included participants who had severe COVID-19 (defined as requiring ICU hospitalization). Across the sampled papers, there were inconsistencies in vaccination within samples: [13,14] both excluded participants with vaccinations; in contrast, [12] included 15 participants who were partially vaccinated (2 of whom were fully vaccinated). No papers collected data on pediatric samples, and all studies excluded participants who were undergoing thromboembolic events or those using anticoagulants.

### 3.3. Population Characteristics

This topic details the characteristics of samples within experimental papers. This includes data on characteristics such as age, sex, ethnicity, and comorbidities of each study’s sample. Given that 3 of the papers reviewed in this article are narrative reviews, they do not have any data on population characteristics. Of the three experiments reviewed, their population was predominantly female at an average of 63%, with [12] having a 61.3% female sample, [13] a 65% female sample, and [14] having a 60.5% female sample. Ref. [13]’s severe COVID-19 cohort was contrastingly 88% male. Sample sizes were smaller due to their conduction at hospitals during the pandemic crisis, with [12] having 62 participants, [13] having 85 participants; however, ref. [14]’s study obtained significantly more participants, with 199 LC patients and 79 controls, likely due to the importance of having a larger sample size when assessing genomes. Across the three studies, mild cases had an average age of 47.4 years, and for [13], severe cases had a median age of 62 years. No studies reported on ethnicities, but all studies reported on common comorbidities of their sample, though most studies did not provide percentages of their afflicted sample. The most common co-morbidities reported were hypertension, diabetes mellitus, obesity, smoking, hypercholesterolemia/hypertriglyceridemia, and heart disease.

### 3.4. Prevalence of Thrombophilic Conditions in Long COVID

This section focuses on data concerning the number of patients enrolled in each study with thrombophilic profiles. It also focuses on whether these populations are at a greater risk of developing LC or of greater severity. The majority of reviewed papers, besides those studying the co-morbidities of LC and thrombophilic diseases, did not include patients with diagnoses of thrombophilic conditions before experimentation. However, many developed or had at minimum, temporary, hypercoagulable states within the timespan of these papers’ experimentations. Thus, the outcomes of these patients may reflect the outcomes of those diagnosed with chronic hypercoagulable conditions. Multiple papers indicated a potential association in patients who exhibited hypercoagulable traits before, during, or shortly following a SARS-CoV-2 infection and the development of LC, with [12]’s results finding that patients with hypercoagulable ROTEMs during hospitalization were reported to be between 17 and 28 times more likely to have an impaired diffusion capacity for carbon monoxide (DLco), a metric commonly seen in LC patients. Furthermore, [12] found in their experiment that individuals with hypercoagulable Rotational Thromboelastometric (ROTEM) profiles whilst sick with COVID-19 or one month post-infection were associated with a greater likelihood of developing LC, which was supported by [9]’s findings noted in their review that those with LC tended to have hypercoagulable ROTEM profiles. Paper [10] also noted in their narrative review elevated levels of certain clotting factors, with 25% to 30% of their sample maintaining elevated levels of D-dimers and increased numbers of endothelial cells (indicating cellular inflammation) in their bloodstream for up to 4 months post-COVID-19.

Some papers investigated the increased risks of developing LC based on genetic factors and co-morbidities. [9]’s review, when focused on patients with diabetes mellitus (DM), an acquired thrombophilic condition, found that these patients were more likely to develop LC, and if so, were at risk for more severe symptoms and worse outcomes. Additionally, this paper noted an elevated level in blood-clotting factors such as tissue factor, factor VII, and Von Willebrand Factor (VWF), placing these patients at a greater risk of forming thromboembolisms and microthrombi. [14]’s study noted that the CC genotype of the Methylenetetrahydrofolate Reductase (MTHFR) gene, held by 49% of their LC cohort, was linked to increased expression of this gene, which may impact its role in regulating the hormone homocysteine (Hcy) [15,16], potentially leading to hypercoagulation. [11]’s review further supports this finding, hypothesizing that, based on LC’s similar vascular pathology to diseases like Type 2 diabetes mellitus (T2DM) and Late Onset Alzheimer’s Disease (LOAD), patients may have an increase in Hcy within their bloodstream (known as hyperhomocysteinemia (HHcy)) due to dysfunctional metabolism of Hcy, which has been shown to lead to pro-thrombotic and pro-inflammatory conditions, such as LC.

### 3.5. Symptom Profiles of Patients with Thrombophilic Conditions and Long COVID

To determine the symptom profiles of those with thrombophilia and LC, data was extracted on topics such as the prevalence of symptoms within a thrombophilic cohort compared to controls, as well as whether these symptoms could be related to the formation of microthrombi. Most papers did not report separately on the thrombophilic patients’ symptoms in LC; however, symptoms across all of the studies’ samples do indicate the formation of microthrombi. [12]’s study found that patients with LC symptoms one month post SARS-CoV-2 infection (36.3% of the original sample) most commonly experienced the symptoms of fatigue, dyspnea, hair loss, anxiety, ageusia, memory issues, and palpitations. Patients exhibiting symptoms 3 months post-infection (22.7% of the original sample) most commonly experienced exertional dyspnea and fatigue. All of these symptoms could be explained by the associated formation of microthrombi in patients with these symptoms, though more research would be necessary to determine a causative relationship. Similarly, [14]’s study reported on the symptoms of LC patients with different alleles of the MTHFR gene, and found that those with the CC genotype, which may be linked to hypercoagulability, were more likely than those with other genotypes to experience the symptoms of hair loss, throat irritation, insomnia, irritability, dyslipidemia, arrhythmia/tachycardia and bradycardia, anxiety, anosmia/hyposmia/parosmia, dizziness, skin itching and spots, hyperglycemia/diabetes, asthma, sweating, myalgias, and muscle weakness, symptoms that concurred with [12]’s findings, and also may indicate a relationship between these symptoms and microthrombi formation.

### 3.6. Potential Mechanistic Pathways of Long COVID and Thrombophilia

This section examines the mechanistic pathways of LC and thrombophilia within our dataset, identifying data that indicated reasonings for the symptom profiles derived from the previous section. We were also looking for scans or tests that could help confirm specific mechanisms, such as evidence of microclots. All papers provided hypotheses on LC’s mechanisms of action. These hypotheses are largely divided into two groups: the increased formation of microthrombi due to elevated levels of pro-coagulants, and the promotion of inflammation post-SARS-CoV-2 infection.

Of the papers that focused on the formation of microthrombi as the mechanism of action for LC, two separate findings supported this theory: first, the increase in pro-coagulants in the bloodstream; and second, the dysfunction or decrease in anti-coagulants within the bloodstream. Collectively, ref. [12]’s study, and refs. [9,10]’s reviews found increases in factors like A10, A20, ALP, MCF, platelet factor IV, VWF, antiphospholipid antibodies, and neutrophil extracellular traps (NETs), all of which have roles in promoting coagulation, leading to a pro-thrombotic state. Ref. [11]’s hypothesis within their narrative review, that the folate-mediated one-carbon metabolism (FOCM) was impaired, leading to a build-up in Hcy, resulting in procoagulative states, is supported by [14]’s study’s findings that the polymorphism of the MTHFR gene into the CC genotype may impair folate metabolism and cause the effects hypothesized in [13]. Furthermore, ref. [13]’s study and ref. [9]’s review found a reduced ability of the body to remove clots, with [13] observing that patients with initial severe COVID-19 maintained hypofibrinolysis beyond 40 days post-infection, and [9]’s review identifying the elevated levels of α-2-antiplasmin and VWF, which were hypothesized to impair normative fibrinolysis. These results suggest a correlation between decreased in anti-coagulant activity and susceptibility to blood clotting, making the body more susceptible to clotting. All of these findings suggest a hypercoagulable state of increased clotting ability with an impaired innate ability to clear said clots, potentially leading to numerous microthrombi formations. This can be seen with [12]’s finding that patients exhibiting hypercoagulability at enrollment, during infection, and post-infection were likely to have less than 80% of their expected DLco, may provide evidence consistent with an association between clotting and an impact on normal respiratory functions.

Additionally, some papers touched on an increase in pro-inflammatory molecules and mechanisms within LC, linked to the onset of clotting and genetic components. [14]’s study found that the AA genotype of the interferon-gamma (IFNG or IFN-γ) gene, which was present in 60% of their LC cohort, led to a lower expression of the cytokine IFN-γ, potentially leading to an impaired immune system, allowing for a higher level of inflammation to occur. [10]’s review identified increased endothelial action and dysfunction, evidenced via an increase in endothelial cells in the bloodstream, and activation of the complement system within LC, relating it to a pro-inflammatory state. These findings may explain the chronic inflammation often associated with LC and its symptoms.

### 3.7. The Effect of Primary (Inherited) Thrombophilia Versus Secondary (Acquired) Thrombophilia on Long COVID Symptoms and Outlook

This section was designed to investigate if there was any difference in secondary and primary thrombophilia in mechanisms or symptomologies when interacting with co-morbidities, in this case, LC. No studies within this review compared primary and secondary thrombophilic conditions’ symptom profiles or outlooks in LC; instead, the majority of papers focused on non-diagnosed participants or participants with a secondary thrombophilic condition, but no paper compared the symptoms, outlooks, or lab results of patients with these different conditions in their paper.

### 3.8. Confounding Variables and Biases of Research

This section primarily focuses on the limitations of the papers included in the dataset to better understand the limitations of the conclusions derived from these results. It also identifies common patterns that may signal institutional barriers, such as a lack of data or newer research fields. The narrative review articles primarily suffered from the limitations of this form of research. Narrative review articles are subject to being easily influenced in their outcomes based on the biases and confounding variables of the studies they rely upon, making them subject to inaccuracies, especially if the research being discussed is preliminary. Additionally, these papers specifically may struggle to draw reliable conclusions due to the lack of research on LC and COVID-19’s interactions with comorbidities at the time the authors were writing their papers. Given that LC is a chronic condition, it is not fully understood to this day, making drawing reliable conclusions difficult.

Of the 3 experimental designs analyzed within this review, two of them were labeled as single-center studies, and all of the studies had small samples obtained from localized areas, reducing the generalizability of these results outside of these small communities. These studies also faced participant attrition, given the nature of these studies requiring frequent follow-ups that posed significant barriers for many. Some studies struggled to conduct their research longitudinally, as seen in [12]’s study, prematurely concluding due to staff shortages in the COVID-19 crisis. Additionally, the majority of these studies did not include fully vaccinated individuals: only 2 participants within [12]’s study had received the full vaccination of 2 doses; thus, their results may differ in comparison to a more vaccinated population. However, these studies did match participants by their age and sex to prevent confounding variables, though this was not done with comorbidities, such as smoking.

### 3.9. Key Takeaways

This section was included to concisely synthesize results for each paper to find commonalities in final conclusions. It was also used as quality control to ensure the results extracted from each paper were accurate and reliable. Multiple papers find evidence of blood clotting factors being linked to the development of LC [11,12,13], from genetic factors (as seen in paper [14]), and acquired conditions, following SARS-CoV-2 infection. Development of COVID-19 has been associated with the development of a hypercoagulative state, and those who go on to develop LC from these infections appear to maintain these conditions, suggesting that this persistent prothrombotic state may cause inflammation, damage, and clotting, leading to LC symptom onset. This is supported by the findings of impaired DLco in LC patients with hypercoagulable ROTEMS [12], and can be seen in the most common symptoms to occur in those with a genotype promoting hypercoagulability being linked to underlying issues of impaired blood circulation due to microthrombi [14].

## 4. Discussion

This paper aimed to investigate whether patients with LC in chronic hypercoagulable states were more likely to develop symptoms related to microthrombi. Multiple papers examined within this systematic review demonstrated evidence of blood clotting factors being linked to the development of LC in both primary and secondary thrombophilic conditions [11,12,13]. Seeing that previous research has observed that the development of COVID-19 leads to a hypercoagulative state during and post-infection, alongside with this review’s finding that those who go on to develop LC from these infections are shown to maintain these hypercoagulable states, there may be a correlation between this persistent prothrombotic state and the inflammatory symptoms and onset of LC. This is supported by the findings of impaired DLco in LC patients with hypercoagulable ROTEMS [12], and can be seen in the most common symptoms to occur in those with a genotype promoting hypercoagulability [14] being linked to underlying issues of impaired blood circulation due to microthrombi.

The results of this review may indicate that patients who have hypercoagulable profiles, or have predispositions to hypercoagulability (both genetic and acquired), may have a higher risk of developing LC, evidenced by being 17–28 times more likely to experience impaired DLco, a common sign associated with LC [12]. Additionally, this evidence suggests that those with predispositions to hypercoagulability may have been more likely to develop symptoms that could be caused by the formation of microthrombi, such as hair loss, arrhythmia, and dizziness. The hypothesis of the increased formation of microthrombi leading to LC symptoms is further supported by this paper’s findings, which identified across multiple papers an increase in coagulation factors such as VWF, ALP, and MCF, as well as an impaired ability to remove clots, also known as hypofibrinolysis. These findings are consistent with past research where a failure to ameliorate the hypercoagulative state brought about by SARS-CoV-2 infection was associated with the development of LC [17,18]. Past research has, on a similar note, also highlighted difficulties in remediating arterial stiffening following COVID-19 infection, particularly in women [19,20]. These findings may also play a role in LC onset and development, and may also explain why the majority of LC patients are female.

An unexpected finding of this paper was the dysregulation of the FOCM system, leading to a surplus of the hormone homocysteine in the blood system, further leading to a coagulative state. This finding suggests that LC may cause damage beyond the induction of pro-thrombotic states, but suggests its mechanisms could potentially lead to dysregulation of the endocrine system as well, though these results should be interpreted with caution. Because LC, as seen in these results, appears to be correlated with hypercoagulable states and microthrombi formation, this present study raises the possibility that patients with a pre-existing predisposition towards clotting via various thrombophilic conditions may be more likely to develop LC, and potentially to a more severe level in their symptomatology. This has been observed in co-morbidities such as diabetes mellitus [4,18] and follows current understandings of thrombophilic conditions as both pro-thrombotic and pro-inflammatory conditions [21]. A possible explanation for this is that thrombophilia, alongside a greater disposition to clotting, has been shown to create chronic inflammation [21,22] due to blood clots and increases in blood clotting factors, which is also observed in COVID-19 patients who go on to develop LC. Prior studies have hypothesized that LC is caused by the prolonged inflammation [23] following COVID-19 infection, so the pre-existing inflammation within thrombophilic patients may make them more susceptible to developing LC. 

This paper’s findings may be useful in suggesting treatments for LC, a chronic disease that, given its novelty and complexity as a highly variable chronic condition and currently has no standard of care.

Due to the findings of increases in blood clotting factors, one potential treatment venue to explore from the findings this paper would be the usage of anticoagulants within LC patients, or even during their initial COVID-19 infection. Though anti-coagulants are used in more severe COVID-19 cases, it may be beneficial to test low-dose anticoagulants, such as acetylsalicylic acid, for mild COVID-19 cases to prevent the formation of microthrombi buildup [24,25] that could potentially lead to LC. Alternatively, anticoagulants could be used once LC symptoms appear, which may prevent the worsening of symptoms due to a compounded formation of microclots. However, recent research has noted that anticoagulants may be ineffective at preventing increases in thrombophilic risk in COVID-19 and thus potentially LC, due to the interaction of immunoinflammatory mechanisms and clotting systems [26]. This finding has been supported in therapeutic evidence, particularly in the use of Tocilizumab to treat patients with COVID-19, which targets the inflammatory mechanisms of COVID-19 instead of the hypercoagulability [27]. Regolo et al.’s 2022 study demonstrated how increased NLR levels, which signify immune system dysfunction and inflammation, acts as a predictor for COVID-19-related mortality and ICU admittance [28], with Regolo et al.’s 2023 study finding inflammatory pathways, including these NLR pathways, to also be predictors of respiratory failure in COVID-19 patients [29], both of these studies’ findings potentially explaining the effectiveness of Tocilizumab’s usage in treating COVID-19. These new investigations could also point to future treatments for LC that are focused on reducing inflammation.

Another implication of this paper would be an increased surveillance of COVID-19 patients’ coagulation profiles. Due to the findings of this paper that hypercoagulability in COVID-19 patients increases the risk of LC development, it may be beneficial for coagulation profiles to be collected to allow practitioners the ability to provide preventative care for LC before its onset. With both patients and doctors having a better understanding of the risks of developing LC, the surveillance of coagulation could reduce the number of patients who develop LC, if sufficient preventative care is achieved, or at least lessen the impact of symptoms. Additionally, this paper’s findings surrounding the dysregulation of the folate-mediated one-carbon metabolism and the subsequent increase in homocysteine levels could potentially be used as an angle for treatment. Due to the fact that increased levels of homocysteine have been linked to vitamin B deficiencies [30], it may be beneficial for LC patients to take supplements of these vitamins to prevent secondary complications.

This paper contributes to existing research by placing an emphasis on the impact of thrombophilia on patients with comorbidities such as LC. Though many thrombophilic conditions are mechanistically understood by themselves, there is abundant room for further progress in determining the effect these conditions may have on a variety of different diseases. This paper also emphasizes the importance of further research into the effects of COVID-19 and LC in systems such as the endocrine system, a topic of research that consists of around 0.5% of completed research on COVID-19 [31]. As seen in this paper, it appears that COVID-19 and Long-COVID can cause damage to the body outside of the blood and lungs, making it an important target of future research.

Recommendations for applications from this paper’s findings come primarily in the treatment of LC patients. It is important to note that this paper’s findings are preliminary research, and extensive testing and research should be completed to validate this paper’s findings before applying any of these applications to patients. This paper’s findings that long-term hypercoagulable profiles existing in COVID-19 make patients more likely to develop LC [12] may suggest a potential use of coagulation profiles for COVID-19 patients to predict the risk of LC onset, allowing for care providers to better assess patient risk and provide preventative care.

In terms of treating LC, this paper’s findings would suggest therapies targeted at the removal of or halting of microthrombi formation, which could be done via the use of anticoagulants or thrombolytic drugs in low doses. Additionally, focus on endocrinal balance and inflammation should also be considered in treating LC, whether through the substitution of vitamins responsible for the breaking of homocysteine, such as vitamin B12 and B6 [30], or the suggestion of dietary and/or lifestyle changes to reduce baseline levels of inflammation.

This paper would also strongly urge an increase in research surrounding LC, given its large impact globally, with an estimated 65 million individuals having LC at some point [32]. COVID-19 is well known for its interaction with various comorbidities [33], and how these comorbidities can greatly influence the severity of illness and the virus’s impact on the coagulatory system [34]. In comparison, LC’s ability to interact with comorbidities is understudied; thus, it is imperative to further research the interactions of comorbidities like thrombophilic conditions with LC. Both topics are relatively unexplored, but pose a significant threat to patients, with several questions remaining unanswered. Topics for future studies include the effect of LC on endocrine systems, the mechanistic differences between primary and secondary thrombophilic conditions and their effects on diseases like COVID-19, and more longitudinal studies investigating the long-term symptoms, mechanisms, and effects of prolonged LC. Additionally, this paper identified a lack of research surrounding LC symptomatology and pathology in more recent years, particularly in vaccinated populations. This would be an interesting topic of research to investigate whether vaccinated LC patients have different symptoms or differences in LC severity. Given that there are currently no studies on patients with thrombophilic conditions before COVID infection, it is not possible to determine whether thrombophilic conditions predispose patients to LC, or whether their thrombophilic conditions are a consequence of the infection.

### Limitations

This paper has potential limitations that may influence its reliability. The sample of papers retrieved is relatively small (n = 6), meaning that the results drawn from this sample are easily influenced by the reliability and credibility of the papers reviewed. This occurred because there is minimal publicly accessible research on the overlapping instances of thrombophilia and LC. In the future, this problem could be solved via an increase in research or by widening the search field in the PubMed database. Additionally, given that we only extracted data from PubMed published articles, this helps authenticate the reliability of the sampled papers, as they were required to undergo a peer-review process before publication. Another limitation of this paper’s methodology of a literature review is its inability to control for tested populations, and due to the small sample size, there is the potential for confounding biases based on geographical location, particularly due to genetic variation. This may make some of the papers’ findings difficult to generalize to wider populations, such as the specific frequency of genotypes of the MTHFR gene. To prevent this aspect from limiting the paper, it would be beneficial to once again sample a larger number of papers over a larger geographic area. A final limitation is the inconsistency within the sample papers of vaccination within their participant pool. Because some papers did include vaccinated participants in their sample, this review may have findings that are unreliable or inconsistent across different populations with different degrees of vaccinations. This could be prevented in the future by incorporating vaccination status into the exclusion criteria while searching.

## 5. Conclusions

There is minimal research into the effects of thrombophilia on the symptoms and mechanisms of LC, which is noteworthy due to thrombophilic patients’ potential increased susceptibility to its development and potential for more severe symptoms. This review highlighted that thrombophilic patients may be at a greater risk of developing LC following initial SARS-CoV-2 infection and are also more likely to experience symptoms related to microthrombi, such as dyspnea, dizziness, and arrhythmia. This is supported by the paper’s findings of elevated levels of clotting factors, hypofibrinolysis, and hormonal and metabolic dysregulation within LC patients, all of which point towards increased clotting ability and prolonged inflammation. Findings point to possible effective strategies for treating LC patients, specifically those with the comorbidity of thrombophilia, via the prevention or reduction in microclots to reduce or remove symptoms. Further research could be conducted on the long-term impacts of different comorbidities on outlook, symptomatology, and mechanisms during LC disease development.

## Figures and Tables

**Figure 1 viruses-17-01315-f001:**
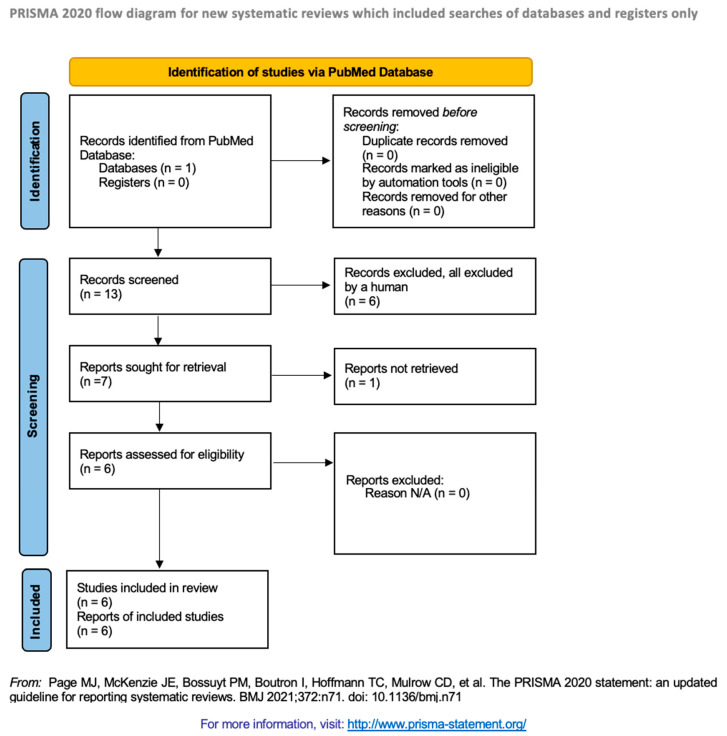
The Preferred Reporting Items for Systematic Reviews and Meta-Analyses (PRISMA) flow chart for our screening protocol [8]. This diagram visualizes the identification and selection process for each of the available articles from identification to final inclusion within review, with (*n* =) representing the number of articles included in each category. Screening resulted in 6 papers (3 narrative reviews and 3 experiments) being included within our systematic review. Each reviewer synthesized data from the papers independently and then agreed upon a final result.

**Table 1 viruses-17-01315-t001:** A table depicting the average number of sentences and the general composition of replies from the LLM ChatGPT version 5.0 to better assist in reproductions.

Category:	Average Number of Sentences Produced by LLM	Primarily Qualitative or Quantitative Data?
Study design and identification	3	Qualitative
Population Characteristics	5	Quantitative
Prevalence of thrombophilic conditions in LC patients	4	Quantitative
Symptom profiles of patients with thrombophilic conditions experiencing LC	5	Qualitative, sometimes Quantitative when articles provided symptom incidences
Potential mechanistic pathways of LC and thrombophilia	7	Qualitative
The effect of primary (inherited) thrombophilia versus secondary (acquired) thrombophilia on LC symptoms and outlook	2	N/A (no data collected)
Confounding and biases of research	5	Qualitative
Key takeaways	4	Equally split between Qualitative and Quantitative

**Table 2 viruses-17-01315-t002:** A table illustrating the quality assessment of each included manuscript according to the Joanna Briggs Institute’s Critical Appraisal Tools. Listed per paper is the checklist utilized, including criterion reached for each question. Percentages were calculated on a points system, with each “YES” response equaling 1 point, each “PARTIAL” response equaling 0.5 points, and each “NO” response equaling 0 points, added up and divided by the largest possible number of points available (also the number of the question). Papers that scored 80% or above were considered strong sources, and studies between 60–79% were considered middle quality, whereas studies that scored under 60% were considered low quality.

Title of Paper	JBI Checklist Used	Criterion Checklist Answers	Percentage of Yes Answers (Counting Partials/Unclears as 0.5 Points)
Hypercoagulable Rotational Thromboelastometry During Hospital Stay Is Associated with Post-Discharge DLco Impairment in Patients with COVID-19-Related Pneumonia	cohort study	1. Were the two groups similar and recruited from the same population? YES 2. Were the exposures measured similarly to assign people to both exposed and unexposed groups? YES 3. Was the exposure measured in a valid and reliable way? YES 4. Were confounding factors identified? YES 5. Were strategies to deal with confounding factors stated? YES 6. Were the groups/participants free of the outcome at the start of the study (or at the moment of exposure)? NO 7. Were the outcomes measured in a valid and reliable way? YES 8. Was the follow up time reported and sufficient to be long enough for outcomes to occur? YES (for short-term) 9. Was follow up complete, and if not, were the reasons to loss to follow up described and explored? PARTIAL 10. Were strategies to address incomplete follow up utilized? PARTIAL 11. Was appropriate statistical analysis used? PARTIAL	77.27%
Persistent hypofibrinolysis in severe COVID-19 associated with elevated fibrinolysis inhibitors activity	cohort study	1. Were the two groups similar and recruited from the same population? YES 2. Were the exposures measured similarly to assign people to both exposed and unexposed groups? YES 3. Was the exposure measured in a valid and reliable way? YES 4. Were confounding factors identified? UNCLEAR 5. Were strategies to deal with confounding factors stated? NO 6. Were the groups/participants free of the outcome at the start of the study (or at the moment of exposure)? NO 7. Were the outcomes measured in a valid and reliable way? YES 8. Was the follow up time reported and sufficient to be long enough for outcomes to occur? YES 9. Was follow up complete, and if not, were the reasons to loss to follow up described and explored? UNCLEAR 10. Were strategies to address incomplete follow up utilized? NO 11. Was appropriate statistical analysis used? YES	63.64%
Mechanisms of endothelial activation, hypercoagulation and thrombosis in COVID-19: a link with diabetes mellitus	Textual assessment: Narrative	1. Is the generator of the narrative a credible or appropriate source? YES 2. Is the relationship between the text and its context explained? (where, when, who with, how) YES 3. Does the narrative present the events using a logical sequence so the reader or listener can understand how it unfolds? YES 4. Do you, as reader or listener of the narrative, arrive at similar conclusions to those drawn by the narrator? YES 5. Do the conclusions flow from the narrative account? YES 6. Do you consider this account to be a narrative? YES	100%
Thrombophilia and Immune-Related Genetic Markers in Long COVID	Cross sectional studies	1. Were the criteria for inclusion in the sample clearly defined? YES 2. Were the study subjects and the setting described in detail? YES 3. Was the exposure measured in a valid and reliable way? YES 4. Were objective, standard criteria used for measurement of the condition? YES 5. Were confounding factors identified? YES 6. Were strategies to deal with confounding factors stated? PARTIAL 7. Were the outcomes measured in a valid and reliable way? YES 8. Was appropriate statistical analysis used? NO	81.25%
SARS-CoV-2 infection and SLE: endothelial dysfunction, atherosclerosis, and thrombosis	Textual assessment: Narrative	1. Is the generator of the narrative a credible or appropriate source? YES 2. Is the relationship between the text and its context explained? (where, when, who with, how) YES 3. Does the narrative present the events using a logical sequence so the reader or listener can understand how it unfolds? YES 4. Do you, as reader or listener of the narrative, arrive at similar conclusions to those drawn by the narrator? YES 5. Do the conclusions flow from the narrative account? YES 6. Do you consider this account to be a narrative? YES	100%
Impaired Folate-Mediated One-Carbon Metabolism in Type 2 Diabetes, Late-Onset Alzheimer’s Disease and Long COVID	Textual assessment: Narrative	1. Is the generator of the narrative a credible or appropriate source? YES 2. Is the relationship between the text and its context explained? (where, when, who with, how) YES 3. Does the narrative present the events using a logical sequence so the reader or listener can understand how it unfolds? YES 4. Do you, as reader or listener of the narrative, arrive at similar conclusions to those drawn by the narrator? YES 5. Do the conclusions flow from the narrative account? YES 6. Do you consider this account to be a narrative? YES	100%

## Data Availability

The exact data extractions can be found in the Figshare database, linked here: https://doi.org/10.6084/m9.figshare.29916908.

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
