# Peer review of "Symptomatology of Long COVID Associated with Inherited and Acquired Thrombophilic Conditions: A Systematic Review"

_viruses, 2025, doi:10.3390/v17101315_

Round 1

Reviewer 1 Report

Comments and Suggestions for Authors

This manuscript by Heath and Li, entitled “Symptomology of Long COVID Associated with Inherited and Acquired Thrombophilic Conditions: A Systematic Review” focused on clinical history of Long Covid as related to both inherited and acquired thrombophilia. Indeed, this topic is interesting, but needs to be better contextualized within the frame of major determinants of vascular damage associated with Covid-19 infection, contributing, together with a dysfunction in hemorheology, to the pathophysiology of  Long Covid. In this regard, both an increase in arterial aging, as well an interplay between inflammation and clotting system, should be also considered. Authors are therefore asked to discuss results coming from their literature’s search, not only as a panel of symptoms of Long Covid and thrombophilic conditions, but also to dissert on emrging mechanisms interacting in the complex pathogenetic chain of thrombophilic burden pending long-term on these patients. Such a vision could help readers to focus on the rationale of using some easy to obtain biomarkers, such as for example neutrophil to lymphocyte ratio  (NLR) and platelet to lymphocyte ratio (PLR), to stratify the prognostic impact of this risk over time. Authors are therefore asked and encouraged to consider further literature’s data to talk about, also to relate them to both Long Covid and Thrombophilic conditions.

            Specific Comments

  1. Evidence that Covid-19 infection is followed by a long-term increase, only partially reversible, in arterial stiffening, was recently shown for the first time by Zanoli et al. (Circ Res 2022). Its meaning was debated to be era proof of either a lasting injury or a prolonged healing (Behrooz L. et al., Circ Res 2022). This topic should be brought to the fore in your systematic review, even in light of very recent data by the very large international Cartesian Study (Bruno R.M. et al., Eur Heart J 2025, ahead of print), confirming that arterial stiffening persists long-term in Covid-19 patients, particularly in women, so emphasizing a gender-related effect.
  2. Inflammatory mechanisms underlying Covid-19 disease were revealed by the derangement of relationships between innate and adaptive immunity (Regolo M. et al., J Clin Med 2022), with complementary interactions between Neutrophils, Lymphocytes and C Reactive Protein in the pathogenetic chain of respiratory failure (Regolo M. et al., J Clin Med 2023). This evidence underscored the concept that the involvement of an interplay between immunoinflammatory mechanisms and clotting system could play a role by either amplifying, or maintaining over time, the thrombophilic risk, despite anticoagulants (Buonacera A. et al., Curr Vasc Pharmacol 2025). Therapeutic evidence supporting this hypothesis comes also from the successful use of Tocilizumab in hospitalized patients with Covid-19 disease (Rezabakhsh A. et al., Arch Acad Emerg Med 2024). So, new targets of therapy look worthy to be considered to face th impactful thrombophilic risk of these patients. In this respect, additional lactate measurement could help to identify prognostic severity (in hospital death or clinical deterioration) in those patients harboring pulmonary embolism (Becattini C. et al., J Thromb Hemost  2024).   .( (

Author Response

Dear Reviewer,

We thank you sincerely for taking the time to review this manuscript. Please find the detailed responses below and the corresponding revisions/ corrections highlighted/in track changes in the re-submitted files.

Response to comments and suggestions for Authors:

Comment 1: Evidence that Covid-19 infection is followed by a long-term increase, only partially reversible, in arterial stiffening, was recently shown for the first time by Zanoli et al. (Circ Res 2022). Its meaning was debated to be era proof of either a lasting injury or a prolonged healing (Behrooz L. et al., Circ Res 2022). This topic should be brought to the fore in your systematic review, even in light of very recent data by the very large international Cartesian Study (Bruno R.M. et al., Eur Heart J 2025, ahead of print), confirming that arterial stiffening persists long-term in Covid-19 patients, particularly in women, so emphasizing a gender-related effect.

Response:  This point is incredibly interesting, and will add depth and nuance to our paper: thank you for raising it. We have included a section discussing arterial stiffening in COVID-19 and its potential effects in the discussion section.

Comment 2: Inflammatory mechanisms underlying Covid-19 disease were revealed by the derangement of relationships between innate and adaptive immunity (Regolo M. et al., J Clin Med 2022), with complementary interactions between Neutrophils, Lymphocytes and C Reactive Protein in the pathogenetic chain of respiratory failure (Regolo M. et al., J Clin Med 2023). This evidence underscored the concept that the involvement of an interplay between immunoinflammatory mechanisms and clotting system could play a role by either amplifying, or maintaining over time, the thrombophilic risk, despite anticoagulants (Buonacera A. et al., Curr Vasc Pharmacol 2025). Therapeutic evidence supporting this hypothesis comes also from the successful use of Tocilizumab in hospitalized patients with Covid-19 disease (Rezabakhsh A. et al., Arch Acad Emerg Med 2024). So, new targets of therapy look worthy to be considered to face th impactful thrombophilic risk of these patients. In this respect, additional lactate measurement could help to identify prognostic severity (in hospital death or clinical deterioration) in those patients harboring pulmonary embolism (Becattini C. et al., J Thromb Hemost  2024).   .( (

Response: We greatly appreciate your addition to our paper with this idea—it adds a greater dimension to the treatment of COVID-19 and LC. We have included a section on this topic within our discussion section where we discuss applications of this review’s findings.

Thank you once more for your thoughtful feedback and consideration of our paper—your suggestions have been essential to bettering our work.

Reviewer 2 Report

Comments and Suggestions for Authors

 Title and Abstract

  • The term “Symptomology” is technically correct but less commonly used than “Symptomatology.”

  • The abstract does not clearly specify the number of studies included or the type of analysis conducted.

 Introduction

  • A brief review of previous studies on the topic is missing.

  • The definition of thrombophilia could be enriched with clinical examples.

  • The objective of the review is clear but not explicitly framed as a research question.

Materials and Methods

  • The time frame of the literature search is not provided.

  • The total number of articles identified, excluded, and included is not indicated.

  • The use of ChatGPT for data extraction is innovative but requires a more rigorous description.

Results

  • There is a lack of summary tables to facilitate comparative reading.

  • Some statements are presented assertively despite being based on preliminary data.

  • The distinction between correlation and causation is not always clearly articulated.

Discussion

  • The therapeutic implications are interesting but remain speculative.

  • A final synthesis of the main contributions is missing.

  • Some claims are too strong relative to the level of evidence available.

Comments on the Quality of English Language

Additional Issues and Recommendations

1. Stylistic and Terminological Consistency

  • Certain terms are used inconsistently—for example, “LC” and “Long COVID” appear interchangeably without a clear pattern.

  • “Hypercoagulable state” and “thrombophilia” are sometimes used as synonyms, but it’s not always clear whether they refer to diagnosed conditions or temporary blood profiles.

Recommendations:

  • Clearly define key terms at the beginning (e.g., LC, thrombophilia, hypercoagulable state) and use them consistently throughout the manuscript.

  • Choose one abbreviation (LC or Long COVID) and apply it uniformly.

2. Handling of Narrative vs. Experimental Sources

  • Narrative reviews are treated on par with experimental studies, despite having different methodological weight.

  • The quality of sources and level of evidence are not explicitly stated.

Recommendations:

  • Clearly distinguish findings derived from narrative reviews versus those from experimental studies.

  • Consider using a quality assessment scale (e.g., GRADE) to classify the strength of evidence.

3. Lack of Quantitative or Statistical Analysis

  • No quantitative synthesis is provided (e.g., aggregated percentages, confidence intervals).

  • Data are presented only descriptively.

Recommendations:

  • If possible, include a table showing the relative frequency of symptoms in thrombophilic versus non-thrombophilic patients.

  • Even a basic comparative analysis (e.g., “X% vs. Y%”) would enhance credibility.

4. Overly Dense Narrative Structure in Some Sections

  • Some paragraphs—especially in the Discussion—are long and dense, making them difficult to read.

Recommendations:

  • Break longer paragraphs into thematic blocks.

  • Use internal subheadings or topic sentences to guide the reader.

5. Absence of a Standalone “Limitations” Section

  • Limitations are discussed within the Discussion but are not highlighted in a dedicated section.

Recommendations:

  • Create a separate “Limitations” section before the Conclusions to emphasize transparency.

6. Potential for Improved Academic Language

  • Some phrases are colloquial or repetitive (e.g., “this paper found that…” used frequently).

Recommendations:

  • Vary sentence structure using more academic expressions such as:

    • “The evidence suggests…”

    • “Findings indicate a potential association…”

    • “This review highlights…”

Author Response

Dear reviewer,

We thank you sincerely for taking the time to review this manuscript. Please find the detailed responses below and the corresponding revisions/ corrections highlighted/in track changes in the re-submitted files.

Comment 1: The term “Symptomology” is technically correct but less commonly used than “Symptomatology.”

            Reply: Thank you so much for raising this point to us. We have replaced all occurences of the word, “Symptomology,” with, “Symptomatology,” within the paper, including within the title.

Comment 2: The abstract does not clearly specify the number of studies included or the type of analysis conducted.

Reply: thank you for noticing this error. We have included these considerations in section 2.2

Comment 3: A brief review of previous studies on the topic is missing.

            Reply: Thank you for raising this subject to our attention. Unfortunately, we were unable to review previous studies on the topic, as this review is discussing a topic with no previously-publsihed review articles, and very few sporadic articles that have discussed individual cases/ mechanisms; this is one of the reasons for the importance of our research. However, we have included an additional section to our introduction explaining this problem to address the lack of review of previous research.

Comment 4: The definition of thrombophilia could be enriched with clinical examples.

Reply: We greatly appreciate this comment, and how it will bring further medical focus into our paper’s review. We have included more content detailing the symptoms of thrombophilic conditions, and about general diagnostic procedures and therapies to better enhance this aspect of our paper.

Comment 5: The objective of the review is clear but not explicitly framed as a research question.

            Reply: thank you for raising this point to us, as it aids in the clarity of our paper. We have added to the very end of the introduction the statement of the explicit research question.

Comment 6: The time frame of the literature search is not provided.

Reply: Thank you for raising this error to us. The timeframe of the literature search has been added to section 2.1.

Comment 7: The total number of articles identified, excluded, and included is not indicated.

            Reply: Thank you for identifying this error. We have included this information in section 2.2.

Comment 8: The use of ChatGPT for data extraction is innovative but requires a more rigorous description.

Reply: We thank you profusely for recognizing the potential roles of Language Learning Models within research fields. To promote visibility, we have included a table in section 2.4.  (Table 1,) detailing the average number of sentences outputted by the LLM, and whether the data was primarily qualitative or quantitative.

Comment 9: There is a lack of summary tables to facilitate comparative reading.

            Reply: Thank you for raising this concern, we feel similarily that the summary tables needed to be easily accessible; thus, a full table of our summary has been published as supplementary materials, under the DOI 10.6084/m9.figshare.29916908.

Comment 10: Some statements are presented assertively despite being based on preliminary data.

Reply: Thank you for raising this concern. We have revised the whole paper, and have changed areas to soften specific assertive phrases.

Comment 11: The distinction between correlation and causation is not always clearly articulated.

            Reply: Thank you for raising this concern. We have reviewed the whole manuscript and have edited the paper to better demonstrate its findings as correlational.

Comment 12: The therapeutic implications are interesting but remain speculative.

Reply: We thank you for taking an interest in the therapeutic implications extracted from our manuscript’s findings. We have included additional sentences when necessary to emphasize these implications as speculative and in need of further research to be useful in medical practice.

Comment 13: A final synthesis of the main contributions is missing.

            Reply: Thank you for addressing this concern. The beginning of the discussion section has been added to synthesize the main and best contributions of the paper’s findings.

Comment 14: Some claims are too strong relative to the level of evidence available.

Reply: Thank you for raising this concern. We have reviewed the whole manuscript and have edited the paper to soften claims to best represent the correlational research.

Additional Comments:

Comment 1: Certain terms are used inconsistently—for example, “LC” and “Long COVID” appear interchangeably without a clear pattern. “Hypercoagulable state” and “thrombophilia” are sometimes used as synonyms, but it’s not always clear whether they refer to diagnosed conditions or temporary blood profiles.

            Reply:  Thank you for raising this concern. Regarding your first point, we have replaced all occurrences of “Long COVID”, save for titles and subtitles, due to its greater recognizability, with the acronym LC. As to your second point, we thank you once again for noting this error in your review. To remedy this, we have reviewed the paper and added phrases like “chronic,” “temporary,” and “long-term” to better distinguish when discussing temporary hypercoagulability and long-term hypercoagulable conditions.

Comment 2: Narrative reviews are treated on par with experimental studies, despite having different methodological weight. The quality of sources and level of evidence are not explicitly stated.

            Reply: Thank you for this point. We have added quality analysis, demonstrated in Table 2, into Section 2.4. We have also edited the paper to better identify within our results when findings have been sourced from experimental or review papers.

Comment 3: No quantitative synthesis is provided (e.g., aggregated percentages, confidence intervals). Data are presented only descriptively.

            Reply: Thank you for raising this concern in your review. Despite our best wishes, we were unable to conduct a quantitative synthesis within our review of the papers, due to the overall lack of quantitative data presented within our reviewed papers’ findings. Additionally, due to the large differences in study design, cohorts, and/or topics of study, combined with the overall lack of quantitative data, we chose to refrain from comparative quantitative analysis between papers to prevent unreliable calculations within our paper. We sincerely thank you for raising this concern, and are remiss we were unable to accommodate it in our revision.

Comment 4: Some paragraphs—especially in the Discussion—are long and dense, making them difficult to read.

            Reply: Thank you for taking notice of this stylistic error: your recognition demonstrates the amount of time you have taken to review our paper, for which we are very grateful. We have further split paragraphs within the discussion to increase their digestibility.

Comment 5: Limitations are discussed within the Discussion but are not highlighted in a dedicated section.

            Reply: Thank you sincerely for raising this concern. A separate limitations section has been included to better distinguish it from the discussion and to increase transparency.

Comment 6: Some phrases are colloquial or repetitive (e.g., “this paper found that…” used frequently).

            Reply: Thank you for raising this concern. We have reviewed the whole manuscript and have edited the paper to include a greater variety of vocabulary that utilizes an academic register.

Thank you once more for your thoughtful feedback and consideration of our paper—your suggestions have been essential to bettering our work.

Reviewer 3 Report

Comments and Suggestions for Authors

In their review, there are only three articles with population data, but the number of patients in each study is not mentioned.

Since there are no studies on thrombophilic conditions prior to COVID infection, it is not possible to determine whether thrombophilic conditions predispose to CL, since it is not possible to determine whether they are a cause or consequence of the infection.

Author Response

Dear reviewer,

We thank you sincerely for taking the time to review this manuscript. Please find the detailed responses below and the corresponding revisions/ corrections highlighted/in track changes in the re-submitted files.

Response to comments and suggestions for Authors:

Comment 1: In their review, there are only three articles with population data, but the number of patients in each study is not mentioned.

Response:  Thank you for bringing up this error within our paper. We have since added the number of participants into the population characteristics section of the Results Section ( which is section 3.3).

Comment 2: Since there are no studies on thrombophilic conditions prior to COVID infection, it is not possible to determine whether thrombophilic conditions predispose to CL, since it is not possible to determine whether they are a cause or consequence of the infection.

Response: We thank you for bringing this point to our attention. To better address this within our paper, we have paraphrased your point and included it in the end of our discussion section.

Thank you once more for your thoughtful feedback and consideration of our paper—your suggestions have been essential to bettering our work.

Round 2

Reviewer 1 Report

Comments and Suggestions for Authors

Major comment

This paper was improved, indeed, but some references and  related iissues on Inflammatory mechanisms underlying  the derangement of relationships between innate and adaptive immunity in COVID-19 disease (Regolo M. et al.,   J Clin Med 2022), with complementary interactions between Neutrophils, Lymphocytes and C Reactive Protein in the pathogenetic chain of respiratory failure (Regolo M. et al., J Clin Med 2023) need to be cited and disserted,, respectively.

Minor Comment

Reference n. 25 includes Christian names, that should be abbreviated.

Author Response

Dear Reviewer,

Thank you kindly for your continued interest in improving our paper. Below are the responses to the corresponding revisions and/or corrections in the track changes of the re-submitted paper:

Comment 1: This paper was improved, indeed, but some references and related issues on Inflammatory mechanisms underlying the derangement of relationships between innate and adaptive immunity in COVID-19 disease (Regolo M. et al.,   J Clin Med 2022), with complementary interactions between Neutrophils, Lymphocytes and C Reactive Protein in the pathogenetic chain of respiratory failure (Regolo M. et al., J Clin Med 2023) need to be cited and disserted, respectively.

We are deeply grateful for your comment on this paper's improvement within the last review round, and for continuing to provide us with new insights to explore within our paper. We have included both studies and their findings within our discussion section and bibliography. We thank you once again for bringing these papers to our attention, as they will surely add greater depth and intrigue to our paper.

Minor comment 1: Reference n. 25 includes Christian names, that should be abbreviated.

Thank you for recognizing this error within our paper. The citation has been correctly formatted to abbreviate Christian names.

Once again, your dedication to reviewing our paper is deeply appreciated, and we thank you deeply for taking an interest in aiding in its revision. 

Reviewer 2 Report

Comments and Suggestions for Authors

Congratulations and thank you for your prompt and thorough responses. The manuscript is now, in my view, ready for publication without further revisions.

Best regards

Author Response

Dear Reviewer,

We thank you sincerely for your kind words and continued support of our paper during the revision process. Attached is the revised manuscript following other reviewers' suggested revisions.

Thank you, once again, for your support in this process.

Round 3

Reviewer 1 Report

Comments and Suggestions for Authors

All concerns have been carefully addressed, allowing to obtain a sensible improvement of the quality of this manuscript.